# ASIR: Steganography for Diffusion Models via Antipodal Sampling and Iterative Recovery

**Yaofei Wang** [1]  **Yufeng Zheng** [1]  **Han Fang** [2]  **Wenzhao Cao** [1]  **Donghui Hu** [1]

## Abstract

Messages embedded in diffusion generation noise suffer from severe attenuation due to denoising and VAE decoding, creating a persistent capacity–robustness trade-off. Identifying that extraction accuracy strictly correlates with the distance between candidate hypothesis images, we propose ASIR, a training-free and provably secure steganography framework for both pixel and latent diffusion models. ASIR introduces two key innovations: (i) Antipodal Sampling, which maximizes signal separation in probability space to enhance distinguishability, and (ii) Iterative Recovery, a paradigm shift that treats extraction as a gradient-based optimization problem to reverse non-linear distortions. Extensive experiments demonstrate that ASIR achieves state-of-the-art performance, embedding up to 65,536 bits (pixel-space) and 16,384 bits (latent-space) with 99% accuracy, while remaining statistically undetectable to deep steganalyzers.

## 1. Introduction

The core objective of steganography is covert communication by concealing the very existence of the message (Simmons, 1984). As digital media evolves, image steganography has shifted from traditional cover-modification methods to generative approaches. Recently, diffusion probabilistic models have emerged as a new frontier in generative steganography due to their high-fidelity generation and ability to model complex data distributions (Ho et al., 2020; Rombach et al., 2022). Unlike traditional methods that modify existing carriers, diffusion-based steganography directly synthesizes stego images by modulating secret information into the random noise of the generation process. In principle,

[1]Hefei University of Technology [2]University of Science and Technology of China. Correspondence to: Donghui Hu <hudh@hfut.edu.cn>.

*Proceedings of the $43^{rd}$ International Conference on Machine Learning*, Seoul, South Korea. PMLR 306, 2026. Copyright 2026 by the author(s).

*Table 1.* Comparison of ASIR with state-of-the-art methods. The payload ($bpp$) is categorized into three levels: High ($bpp \geq 1$), Medium ($0.05 \leq bpp < 1$), and Low ($bpp < 0.05$).

| Method | Pixel-domain | Latent-domain | Payload | Security |
|---|---|---|---|---|
| StegaDDPM | ✔ | ✗ | High | ✔ |
| LDStega | ✗ | ✔ | Medium | ✗ |
| GRDH | ✗ | ✔ | Medium | ✔ |
| MDDM | ✔ | ✔ | Low/Medium | ✔ |
| Diffusion-Stego | ✔ | ✔ | High | ✗ |
| Pulsar | ✔ | ✗ | Medium | ✔ |
| GS | ✗ | ✔ | Low | ✔ |
| **ASIR (Ours)** | ✔ | ✔ | **High** | ✔ |

this avoids the statistical traces left by cover modifications, offering new opportunities for covert communication.

Existing diffusion-based steganography schemes primarily fall into two categories. The first category embeds messages during intermediate denoising steps (Peng et al., 2023; 2024; Jois et al., 2024). For example, (Peng et al., 2023) proposed StegaDDPM; however, its embedding/inversion pipeline bypasses the final denoising step, which deviates from mainstream diffusion sampling and makes the scheme brittle under practical generation and post-processing. Pulsar (Jois et al., 2024) addresses this issue by operating within the standard sampling procedure and improving robustness, but it comes at the cost of a relatively low embedding rate. The second category embeds messages in the initial noise. Methods such as GRDH (Hu et al., 2024), GS (Yang et al., 2024b) and MDDM (Xu et al., 2025) rely on powerful inversion techniques to recover the initial noise for high-precision extraction; however, the limited mapping space of the initial noise constrains embedding capacity. To break this bottleneck, Diffusion-Stego (Kim et al., 2025) attempts to increase capacity via alternative message mapping schemes, but such non-standard mappings may disrupt the noise distribution and introduce security vulnerabilities. Table 1 showcases the performance of all aforementioned methods regarding model compatibility, Effective Payload, and security. In contrast, ASIR stands out as the unique solution satisfying all criteria.

Current methods have struggled to simultaneously achieve high capacity and high extraction accuracy while preserving

security. We argue that the core difficulty stems from the severe distortion of the embedded signal caused by non-linear transformations along the generation pipeline.

A key empirical finding in our study is that extraction accuracy is strongly correlated with the distance between the candidate hypothesis images (or trajectories) corresponding to different embedded messages: when these hypotheses become too close after the final denoising step (and, for LDMs, VAE decoding), reliable decoding becomes intrinsically difficult. Existing approaches effectively address this issue only through partial workarounds (e.g., adding redundancy or restricting the embedding space), which inevitably compromises capacity, compatibility, or security.

Motivated by the above observation, we propose ASIR (Antipodal Sampling and Iterative Recovery), a training-free and provably secure steganography framework for diffusion models. To tackle non-linear distortion, ASIR combines proactive defense and reactive correction. First, Antipodal Sampling maximizes the separation between candidate signals in probability space during embedding, providing a more distinguishable basis for decoding. Second, Iterative Recovery leverages the differentiability of the diffusion generation pipeline to cast decoding as a gradient-based optimization problem, thereby correcting errors induced by non-linear distortions.

Our main contributions are summarized as follows:

- We propose ASIR, a novel training-free steganography framework with provable security, compatible with both pixel-space and latent-space diffusion models.

- We introduce Antipodal Sampling, an embedding strategy that maximizes the separation between candidate noises to improve message distinguishability.

- We develop Iterative Recovery, a gradient-based optimization method for message extraction that improves accuracy over direct comparison without sacrificing embedding capacity.

- We demonstrate through extensive experiments that ASIR achieves a superior capacity–accuracy performance while remaining secure against advanced steganalysis tools and preserving image quality.

## 2. Related work

Steganography aims at covert communication by concealing the very existence of the message (Simmons, 1984). Existing image steganography methods can be broadly grouped into cover-modification and generative paradigms.

**Cover-Modification Steganography.** Early methods, such as LSB (Mielikainen, 2006), embed messages by mod-

ifying pixel values. Later approaches improve security by combining coding schemes (e.g., SPC (Li et al., 2020), STC (Filler et al., 2011)) with distortion minimization (e.g., HUGO (Pevnỳ et al., 2010), SUNIWARD (Holub et al., 2014)) and learned embedders (e.g., HiDDeN (Zhu et al., 2018), SteganoGAN (Zhang et al., 2019)). However, these modification-based approaches inevitably introduce pixel-level perturbations, leaving traceable artifacts for steganalysis.

**Generative Steganography.** Generative methods avoid explicit cover modification by synthesizing images with embedded messages. Early works primarily leveraged GANs (Goodfellow et al., 2014) and Flow models (Kingma & Dhariwal, 2018). Representative approaches embed messages through distinct mechanisms, such as feature map fusion in GSN (Wei et al., 2022), disentangled representations in IDEAS (Liu et al., 2022), and bijective mappings in S2IRT (Zhou et al., 2022). However, a significant drawback of these methods is the high computational overhead incurred by the necessity of model retraining. Although training-free GAN schemes (Yang et al., 2023) have been proposed to alleviate this issue, their overall performance remains fundamentally limited by the generation quality and controllability of the underlying GANs.

**Diffusion-based Steganography.** Recently, Diffusion Models have garnered significant attention due to their superior generation fidelity. Current diffusion-based steganography can be broadly categorized into two streams: embedding messages during intermediate denoising steps (Jois et al., 2024; Peng et al., 2023; 2024), or mapping messages directly to the initial noise distribution (Hu et al., 2024; Kim et al., 2025; Xu et al., 2025; Yuan et al., 2025), typically relying on deterministic samplers (Lu et al., 2025) or inversion techniques (e.g., EDICT (Wallace et al., 2023)). Despite their promise, these methods often lack scalability and struggle to achieve high embedding capacity without compromising security or accuracy. Note that recent techniques involving concealing full images within carrier images (Yu et al., 2023; Yang et al., 2024a; Yan et al., 2025) fundamentally differ from the binary message embedding discussed in this work.

## 3. Preliminary

Diffusion probabilistic models generate high-fidelity data by reversing a gradual noise-adding process. In the foundational Denoising Diffusion Probabilistic Models (DDPM) (Ho et al., 2020), the generative process is defined as a reverse Markov chain. Starting from standard Gaussian noise $\mathbf{x}_T \sim \mathcal{N}(\mathbf{0}, \mathbf{I})$, the model iteratively denoises the data to reconstruct $\mathbf{x}_0$. Crucially, each step in this reverse process introduces stochasticity via a random noise term $\mathbf{z}$. The

sampling update rule is formulated as:

$$\mathbf{x}_{t-1} = \underbrace{\frac{1}{\sqrt{\alpha_t}}\left(\mathbf{x}_t - \frac{\beta_t}{\sqrt{1-\bar{\alpha}_t}}\boldsymbol{\epsilon}_\theta(\mathbf{x}_t, t)\right)}_{\text{Deterministic Prediction } \boldsymbol{\mu}_\theta} + \underbrace{\sigma_t \mathbf{z}}_{\text{Stochastic Carrier}}$$

(1)

where $\boldsymbol{\epsilon}_\theta$ is the noise predicted by the neural network, $\alpha_t$ and $\beta_t$ are schedule parameters, and $\mathbf{z} \sim \mathcal{N}(\mathbf{0}, \mathbf{I})$ is the variance noise. This explicitly injected Gaussian noise $\mathbf{z}$ ensures the diversity of the generated samples.

To reduce computational costs, Latent Diffusion Models (LDMs) (Rombach et al., 2022) shift this process into a compressed latent space. An encoder $\mathcal{E}$ maps the input image $\mathbf{I}$ to a latent representation $\mathbf{x}_0 = \mathcal{E}(\mathbf{I})$. Consequently, the diffusion process (Eq. 1) is performed on these latent vectors $\mathbf{x}_t$. The final image is reconstructed via a decoder $\hat{\mathbf{I}} = \mathcal{D}(\mathbf{x}_0)$.

# 4. Motivation

## 4.1. Signal Attenuation in Generative Channels

Embedding secret messages into the diffusion process effectively treats the generative model as a noisy communication channel. Existing methods typically inject messages by modulating the sampling noise at a specific timestep (Peng et al., 2023; 2024; Jois et al., 2024). However, the subsequent reverse diffusion process acts as a rigorous denoiser. As the generation iterates from $t$ to $0$, the high-frequency perturbations introduced by the secret message are progressively smoothed out. This attenuation is exacerbated in Latent Diffusion Models (LDMs), where the VAE decoder further functions as a low-pass filter, significantly shrinking the magnitude of the embedded signal in the final pixel space. Consequently, a randomly encoded signal often becomes indistinguishable from natural image variations, leading to decoding failures.

## 4.2. The Correlation between Signal Separation and Accuracy

To quantify the impact of this attenuation, we analyze the geometric relationship between the embedded latent codes and the decodability of the generated images. We define the **Signal Separation**, denoted as $\Delta$, as the per-pixel absolute difference between two generated image instances conditioned on opposing message bits (e.g., bit '0' vs. bit '1'):

$$\Delta_{h,w} = |\mathbf{x}_0^{(m=0)}(h,w) - \mathbf{x}_0^{(m=1)}(h,w)| \quad (2)$$

where $\mathbf{x}_0^{(m)}$ represents the final output generated with message $m$. As visualized in Figure 1, we observe a critical empirical law: *Extraction accuracy is strictly positively correlated with Signal Separation.*

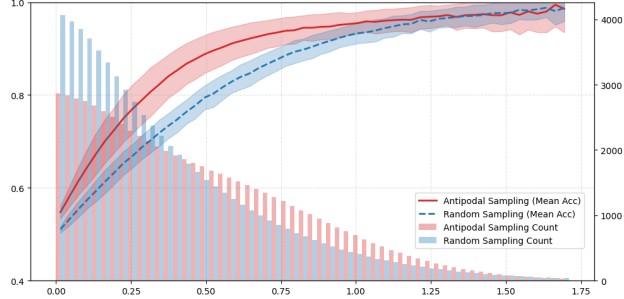

*Figure 1.* **Impact of Signal Separation on Extraction Accuracy.** The x-axis represents the per-pixel Signal Separation ($\Delta_{h,w} = |\mathbf{x}_0^{(0)}(h,w) - \mathbf{x}_0^{(1)}(h,w)|$) between images generated for bit '0' and '1'. The bar chart (right y-axis) shows the distribution of $\Delta_{h,w}$. The line plot (left y-axis) shows that accuracy increases monotonically with $\Delta_{h,w}$.

**The Failure of Random Sampling (Blue Region).** Standard steganography methods rely on random sampling to map messages to noise vectors (e.g., mapping '0' and '1' to two independent Gaussian vectors $\mathbf{z}_a, \mathbf{z}_b \sim \mathcal{N}(0, \mathbf{I})$). Since candidate noise vectors are sampled independently, their probabilities lack maximum separation constraints, causing them to be close. As shown by the blue histogram in Figure 1, the resulting distribution of Signal Separation $\Delta$ is heavily skewed towards zero. This indicates that for the vast majority of random samples, the generated image pairs for '0' and '1' are geometrically too close to withstand the signal attenuation described above, leading to a collapse in extraction accuracy.

**The Necessity of Antipodal Sampling (Red Region).** To ensure robustness, the embedding strategy must proactively maximize $\Delta$ to survive the denoising process. This motivates our proposed **Antipodal Sampling**. Instead of relying on chance, we enforce the candidate noise vectors to be diametrically opposite in the probability space. As shown by the red curve in Figure 1, this strategy effectively shifts the distribution of $\Delta$ towards the high-separation regime (the "Safe Zone"), guaranteeing that the generated outputs remain distinguishable even after significant channel attenuation.

# 5. The Proposed ASIR Framework

## 5.1. Overall Framework

ASIR is a training-free and provably secure steganography framework compatible with both pixel-space and latent-space diffusion models. The core principle of ASIR is to embed a secret message into the variance noise component of a single reverse diffusion step, meticulously preserving the noise's statistical properties to ensure security. The framework is built upon two key innovations: Antipodal Sampling, a novel embedding strategy that maximizes message distinguishability, and Iterative Recovery, a ro-

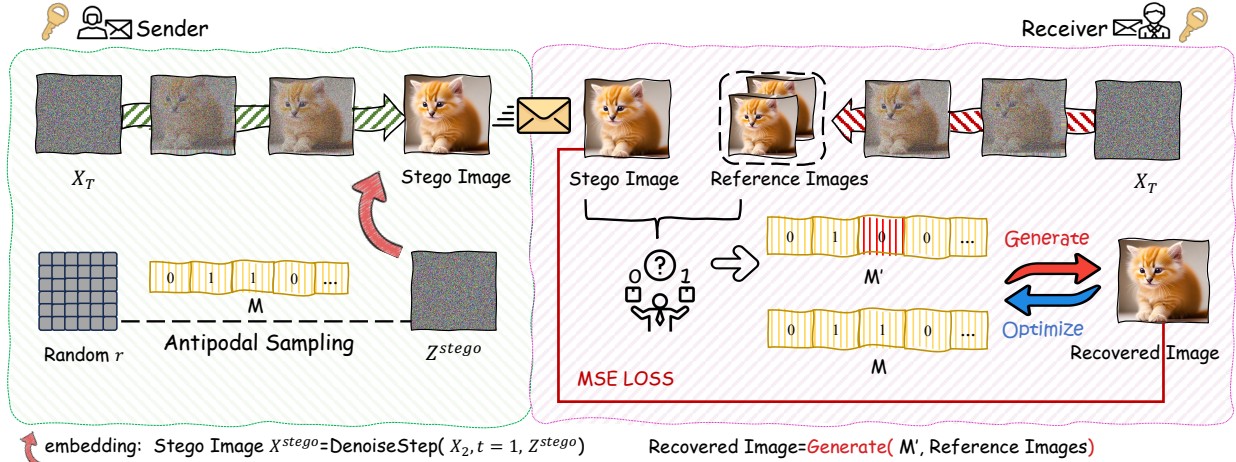

*Figure 2.* Workflow of the ASIR Framework. (a) Embedding phase: The sender modulates variance noise via Antipodal Sampling at the penultimate diffusion step ($t = 1$) to encode secret messages; (b) Extraction phase: The receiver performs preliminary decoding through direct Euclidean comparison and refines the output via Iterative Recovery optimization.

bust extraction mechanism that achieves high accuracy. An overview of the entire process is illustrated in Figure 2.

The operational principle of ASIR hinges on synchronized generation and comparative decoding, orchestrated via a shared secret key $k_s$. This key ensures that both sender and receiver follow an identical, pseudo-random diffusion trajectory up to the point of embedding. The process unfolds as follows:

- **Embedding by Sender:** During the penultimate denoising step ($t = 1$), the sender embeds the secret message **m**. This is not done by adding unstructured noise, but by modulating the variance noise using our proposed Antipodal Sampling technique. This structured modulation produces an intermediate state $X_1^{\text{stego}}$, which then undergoes the final denoising step to yield the stego image $X_0^{\text{stego}}$.

- **Decoding by Receiver:** The receiver, using the same key $k_s$, leverages the synchronized process to decode the message. The initial message estimate is then found by identifying which reference image is most similar to the received stego image $X_0^{\text{stego}}$ based on Euclidean distance. This provides a preliminary prediction, which is subsequently refined by our Iterative Recovery module to achieve high accuracy.

### 5.2. Antipodal Sampling for Embedding

A naive approach to embedding, like mapping message bits to independently sampled noise vectors (Jois et al., 2024), is fundamentally flawed. Its reliance on chance means the noise samples are not guaranteed to be distinguishable and can be statistically close, leading to a high bit error rate after denoising and quantization .

To solve this, we introduce Antipodal Sampling, which guarantees maximum separation between signals. Instead of drawing independent samples, our method controls the relative positioning of candidate noise vectors. We generate a single shared random base and then derive maximally separated "antipodal" points from it in a structured, message-dependent manner . This process unfolds in two stages:

**Maximal Separation in Probability Space** We begin in the uniform probability interval $[0, 1]$. To embed a single bit—a choice between two states—we need to select two points in this interval to represent '0' and '1'. To make them as distinct as possible, we should place them at opposite ends of the interval. On a circular space (which the modulo operation effectively creates), the point most distant from any value $r$ is located halfway around the circle, at a distance of 0.5.

Therefore, we first use a shared key $k_s$ to generate a single pseudo-random number $r \sim \mathcal{U}[0, 1]$, which serves as a random anchor. The two candidate probability values, $p(0)$ for bit '0' and $p(1)$ for bit '1', are then defined as:

$$p(0) = r \tag{3}$$
$$p(1) = (r + 0.5) \pmod 1 \tag{4}$$

This construction guarantees that regardless of the random value of $r$, the two probabilities $p(0)$ and $p(1)$ are always separated by a distance of exactly 0.5, ensuring maximal separation on the unit interval. The sender then selects one of these probabilities based on the secret message bit $m \in \{0, 1\}$ to get the final probability $p(m)$.

This principle naturally extends to higher-capacity embedding. Let $R$ denote the number of candidate states. There are $R$ possible messages (corresponding to $\log_2 R$ bits). To

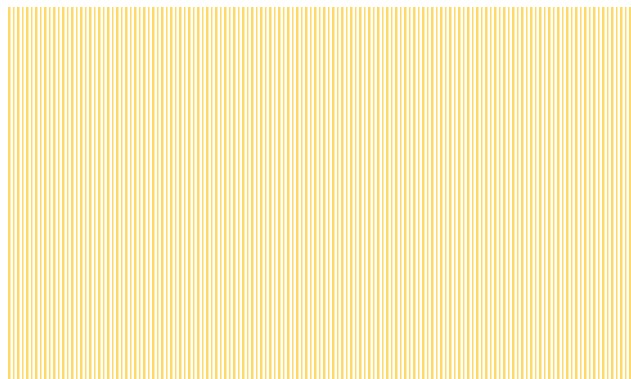

*Figure 3.* Antipodal Sampling Noise Generation Mechanism (R=2). (a) Probability space: Maximally separated candidate probabilities ($\Delta = 0.5$) generated from shared random anchor r; (b) Noise transformation: Gaussian noise samples $Z^{stego}$ derived through inverse CDF mapping.

space them equally, we divide the unit interval into $R$ segments, resulting in a separation of $1/R$ between adjacent message representations.

**Transformation to Gaussian Noise**  The essence of message embedding is to convert a uniform distribution into a standard Gaussian distribution. To embed the message within the variance noise inherent to the denoising process, we leverage the *inverse transform sampling* method. This requires mapping the uniform variable $p(m)$ to a sample $Z^{stego}$ from the standard Gaussian distribution $\mathcal{N}(0,1)$.

The Cumulative Distribution Function (CDF) of the standard Gaussian distribution, $F(z) = P(Z \leq z)$, is given by:

$$F(z) = \int_{-\infty}^{z} \phi(t)dt = \int_{-\infty}^{z} \frac{1}{\sqrt{2\pi}} \exp\left(-\frac{t^2}{2}\right) dt \quad (5)$$

The Inverse CDF (ICDF), maps a probability $p \in [0,1)$ back to the domain $z \in \mathbb{R}$. We use $p(m)$ as a probability:

$$Z^{stego} = F^{-1}(p(m)) = \inf\{z \in \mathbb{R} \mid F(z) \geq p(m)\} \quad (6)$$

By construction, applying the ICDF to a uniformly distributed variable $p(m)$ yields a sample $Z^{stego} \sim \mathcal{N}(0,1)$ that is deterministically linked to the message $m$ and pseudo-random number $r$. By maximizing the interval between pseudo-random number $r$ sampled from a uniform distribution, Antipodal Sampling ensures the corresponding noise samples $Z^{stego}$ are maximally separated under the Gaussian measure, directly tackling the distinguishability problem inherent in purely random sampling methods like Pulsar.

The complete procedure for generating the stego image incorporating $Z^{stego}$ is detailed in Algorithm 1. Figure 3 provides an illustrative example for the case $R = 2$ (representing 1 bit), demonstrating the separation principle.

---

**Algorithm 1** ASIR.Encode

---

**Require:** Shared key $k_s$, Number of candidate states $R$, Message matrix $\mathbf{M}$ of size $H \times W$ with elements in $\{0, ..., R-1\}$
**Ensure:** Stego image $X_0^{\text{stego}}$
 1: $\mathbf{r} \leftarrow \text{Sample}(k_s, \mathcal{U}(0,1)^{H \times W})$
 2: $X_T \leftarrow \text{Sample}(k_s, \mathcal{N}(0, \mathbf{I}))$ {Initialize noise image}
 3: **for** $t \leftarrow T$ **down to** 2 **do**
 4: $\quad X_{t-1} \leftarrow \text{DenoiseStep}(X_t, t, \text{Sample}(k_s, \mathcal{N}(0, \mathbf{I})))$
 5: **end for**
 6: $\mathbf{P_m} \leftarrow (\mathbf{r} + \mathbf{M}/R) \pmod 1$ {Calculate probability matrix}
 7: $Z_1^{\text{stego}} \leftarrow F^{-1}(\mathbf{P_m})$ {Generate structured noise via Inverse CDF}
 8: $X_1^{\text{stego}} \leftarrow \text{DenoiseStep}(X_2, t = 1, Z_1^{\text{stego}})$
 9: $X_0^{\text{stego}} \leftarrow \text{DenoiseStep}(X_1^{\text{stego}}, t = 0, \text{noise} = \text{None})$
10: **return** $X_0^{\text{stego}}$

---

**Algorithm 2** ASIR.Decode

---

**Require:** Shared key $k_s$, number of candidate states $R$, stego image $X_0^{\text{stego}}$
**Ensure:** Recovered message matrix $\mathbf{M}_{\text{final}}$
 1: $\mathbf{r} \leftarrow \text{Sample}(k_s, \mathcal{U}(0,1)^{H \times W})$
 2: $X_T \leftarrow \text{Sample}(k_s, \mathcal{N}(0, \mathbf{I}))$ {*Synchronized Reconstruction*}
 3: **for** $t \leftarrow T$ **down to** 2 **do**
 4: $\quad X_{t-1} \leftarrow \text{DenoiseStep}(X_t, t, \text{Sample}(k_s, \mathcal{N}(0, \mathbf{I})))$
 5: **end for**
 6: **for** $j \leftarrow 0$ **to** $R-1$ **do**
 7: $\quad P_j \leftarrow (\mathbf{r} + j/R) \pmod 1$
 8: $\quad Z_1^j \leftarrow F^{-1}(P_j)$
 9: $\quad X_1^j \leftarrow \text{DenoiseStep}(X_2, 1, Z_1^j)$
10: $\quad X_0^j \leftarrow \text{DenoiseStep}(X_1^j, 0, \text{noise=None})$
11: $\quad \text{RefImages}[j] \leftarrow X_0^j$
12: **end for**
13: Let $\mathbf{M}_{\text{init}}$ be an empty $H \times W$ matrix
14: **for** each position $(h, w)$ **do**
15: $\quad \mathbf{M}_{\text{init}}(h, w) \leftarrow$
16: $\quad\quad\quad \arg\min_j \left| X_0^{\text{stego}}(h, w) - \text{RefImages}[j](h, w) \right|$
17: **end for**
18: $\mathbf{M}' \leftarrow \text{int2float}(\mathbf{M}_{\text{init}})$ {Initialize continuous message matrix}
19: **for** $k \leftarrow 1$ **to** OptimizationSteps **do**
20: $\quad \mathcal{L}(\mathbf{M}') \leftarrow \left\| X_0^{\text{stego}} - \text{Generate}(\mathbf{M}', X_2, r_{\text{matrix}}) \right\|_2^2$
21: $\quad \mathbf{M}' \leftarrow \mathbf{M}' - \gamma \nabla_{\mathbf{M}'} \mathcal{L}(\mathbf{M}')$
22: **end for**
23: $\mathbf{M}_{\text{final}} \leftarrow \text{round}(\mathbf{M}')$
24: **return** $\mathbf{M}_{\text{final}}$

---

## 5.3. Iterative Recovery for Extraction

**The Limitation of Direct Comparison** While Antipodal Sampling provides a strong initial signal separation, the preliminary decoding via direct Euclidean comparison is limited. This one-shot method ignores the complex, non-linear distortions from the final denoising step and VAE decoding, resulting in a non-negligible bit error rate. While conventional Error-Correcting Codes (ECC) can mitigate such errors, they fundamentally trade capacity for robustness—a compromise we seek to avoid .

**Message as a Differentiable Parameter** To overcome this limitation without sacrificing capacity, we introduce Iterative Recovery, a method that reframes the decoding task from a simple comparison problem into a gradient-based optimization problem. The core insight is to treat the secret message not as a discrete set of bits to be guessed, but as a continuous, differentiable parameter that can be optimized.

Since the diffusion model's generation process is end-to-end differentiable, we can directly compute the gradient of the final image with respect to the input message. This allows us to "ask" the model: "How should I change my current message estimate to make the reconstructed image look more like the received stego image?"

**The Optimization Process** The process begins with the initial message matrix $\mathbf{M}_{\text{init}}$, obtained from the pixel-wise direct comparison method. We then aim to find an optimal message matrix $\mathbf{M}'$ that minimizes the following loss function—the squared L2 distance between the received stego image $X_0^{\text{stego}}$ and a reconstructed image, which is generated using the current message matrix estimate $\mathbf{M}'$:

$$\mathcal{L}(\mathbf{M}') = \left\| X_0^{\text{stego}} - \text{Generate}(\mathbf{M}', X_2, r_{\text{matrix}}) \right\|_2^2 \quad (7)$$

The helper function $\text{Generate}(\mathbf{M}', X_2, r_{\text{matrix}})$ encapsulates the differentiable forward process, taking the current continuous message matrix estimate $\mathbf{M}'$ and generating a corresponding reconstructed image. Leveraging the differentiability of the generative pipeline, we iteratively refine the entire message matrix $\mathbf{M}'$ using gradient descent:

$$\mathbf{M}' \leftarrow \mathbf{M}' - \gamma \nabla_{\mathbf{M}'} \mathcal{L}(\mathbf{M}') \quad (8)$$

where $\gamma$ is the learning rate. This optimization process effectively inverts the generation pipeline, treating the entire secret message matrix as a free parameter to be discovered. The process continues until the loss converges, at which point the optimized continuous matrix $\mathbf{M}'$ is rounded to the nearest integers to yield the final, highly accurate message matrix $\mathbf{M}_{\text{final}}$. The entire decoding procedure, combining the preliminary comparison and this iterative refinement, is formalized in Algorithm 2.

In practice, $\mathbf{M}'$ is parameterized as learnable logits optimized via differentiable relaxation techniques (Straight-Through Estimator for $R = 2$, Gumbel-Softmax for $R > 2$), enabling gradient flow through the discrete message selection. The function Generate efficiently constructs the reconstructed image by interpolating between pre-computed reference images at the $X_1$ level, followed by the final denoising step.

## 5.4. Security Analysis

The security of ASIR rests on a fundamental distributional equivalence: the stego noise $Z^{\text{stego}}$ follows the exact same distribution $\mathcal{N}(\mathbf{0}, \mathbf{I})$ as the natural sampling noise.

**Proposition 5.1.** *For any message $m$ and any shared key $k_s$, the marginal distribution of the Antipodal Sampling noise satisfies $Z^{\text{stego}} \sim \mathcal{N}(\mathbf{0}, \mathbf{I})$.*

*Proof sketch.* First, the shared anchor $r$ is generated by a CSPRNG keyed by $k_s$ and is therefore computationally indistinguishable from $\mathcal{U}[0, 1)$. Next, the message-dependent shift $p(m) = (r + m/R) \bmod 1$ preserves uniformity, since adding a constant modulo 1 to a uniform variable inherently remains uniform. Applying the standard normal ICDF to $p(m) \sim \mathcal{U}[0, 1)$ deterministically yields $Z^{\text{stego}} \sim \mathcal{N}(\mathbf{0}, \mathbf{I})$. Because the transition kernel $P(\mathbf{x}_{t-1}|\mathbf{x}_t)$ depends on the noise strictly through its distribution, the stego and cover generation processes are identically distributed. Consequently, the adversary's distinguishing advantage reduces entirely to breaking the underlying CSPRNG.

## 6. Experiments

In this section, we conduct a comprehensive set of experiments to evaluate the proposed ASIR framework. Specifically, we evaluate ASIR along three dimensions: (1) core performance in terms of steganographic capacity, extraction accuracy, and security; (2) comparison against state-of-the-art baselines; and (3) ablation studies that isolate the contributions of Antipodal Sampling and Iterative Recovery.

### 6.1. Experimental Setup

**Models.** For pixel-space experiments, we use pre-trained DDPM models from Hugging Face, including `ddpm-church-256` (Google, 2022c), `ddpm-bedroom-256` (Google, 2022a), and `ddpm-celebahq-256` (Google, 2022b). For latent-space experiments, we use Stable Diffusion v1.5 and v2.1 (Rombach et al., 2022). Our implementation is built upon the `diffusers` library (von Platen et al., 2022), utilizing the DDIM scheduler (Song et al., 2020) for efficient sampling.

**Hyperparameters.** We employ differentiable relaxation

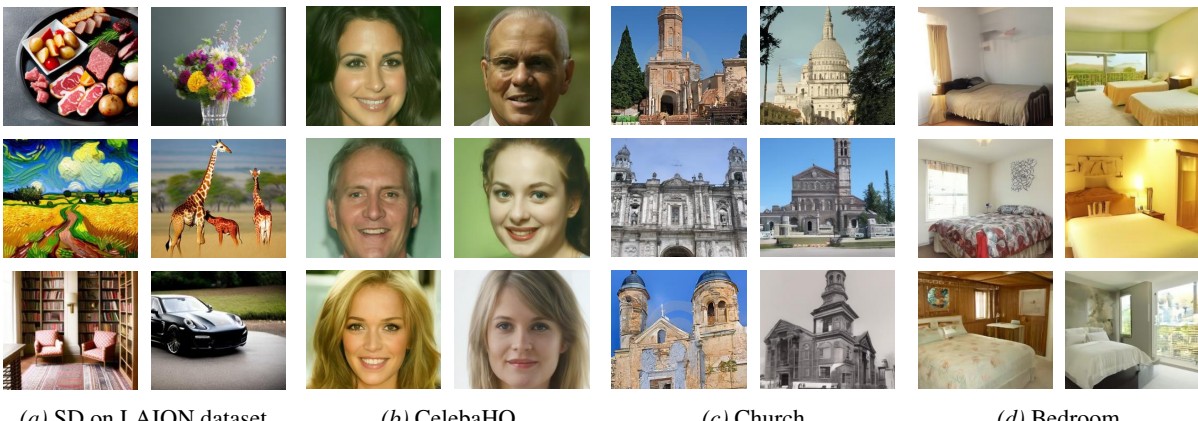

*(a)* SD on LAION dataset       *(b)* CelebaHQ       *(c)* Church       *(d)* Bedroom

*Figure 4.* Showcase of stego images generated by ASIR across different datasets (a-d). Despite embedding payloads ranging from 1,000 to 100,000 bits, the generated samples exhibit high diversity and visual realism, confirming that our method preserves perceptual quality without introducing visible artifacts.

for discrete optimization: a Straight-Through Estimator for binary embedding ($R = 2$) and Gumbel-Softmax ($t = 1.0$) for multi-state embedding ($R > 2$). Learnable logits are initialized from a random normal distribution and updated via the Adam optimizer (lr $= 0.04$). The optimization runs for a maximum of 2,000 iterations, utilizing a sliding-window early stopping strategy. We use a scaled Mean Squared Error (MSE) loss to enhance gradient signals for faster convergence. Finally, the embedded message is recovered by applying an argmax operation to the optimized logits.

All experiments were conducted on a system with an NVIDIA GeForce RTX 4090 GPU (24 GB VRAM), running Ubuntu 22.04.5 LTS. Unless otherwise stated, we evaluate the performance using 1,000 randomly generated samples for each experimental setting.

### 6.2. Performance of ASIR

**Image Quality.** We evaluate perceptual imperceptibility both qualitatively and quantitatively. Qualitatively, Figure 4 demonstrates that ASIR maintains high visual realism without visible artifacts. Quantitatively, we employ the Fréchet Inception Distance (FID) (Heusel et al., 2017) to measure the distributional similarity between ASIR-generated stego images and cover images from standard datasets, including COCO (Lin et al., 2014), CelebAHQ (Karras et al., 2017) ,and LSUN (Church and Bedroom) (Yu et al., 2015). Detailed numerical comparisons are provided in Appendix B.1. The results show a negligible FID difference between cover and stego images. This confirms that the embedding process does not compromise image quality, which remains primarily determined by the underlying diffusion backbone.

**Capacity and Extraction Accuracy.** We utilize Bit Extraction Accuracy (Acc) as the primary metric to quantify the fidelity of the recovered secret message. While the foun-

dational framework (Section 5.1) was introduced with candidate states ($R = 2$), the underlying Antipodal Sampling principle is inherently scalable. To transcend the capacity limitations of fixed embedding schemes, we extend the framework using an Adaptive Embedding Strategy governed by signal distance constraints. Formally, let $R$ denote the number of candidate states, yielding $\log_2 R$ bits per unit. To maximize entropy utilization, we construct hierarchical candidate pools (e.g., $\mathcal{CP} = \{R \mid R \in \{2, 3, 4\}\}$) and employ a "greedy fallback" mechanism: prioritizing the highest order $R$, the system calculates the minimum signal distance among candidates. If this distance falls below a threshold $\tau$, it iteratively reduces $R$ (e.g., $4 \rightarrow 3 \rightarrow 2$); if the distance remains insufficient, the region is skipped. Extensive experiments with various pools and thresholds ($\tau \in [0.0, 0.6]$, see Table 2) yield a critical insight: ASIR effectively decouples capacity scaling from accuracy degradation. By expanding the candidate pool, the framework achieves exponential capacity growth—surpassing 100,000 bits—while consistently maintaining extraction accuracy above 95%, demonstrating exceptional scalability for high-density steganography.

**Security Against Steganalysis.** The security proof of ASIR is provided in Section 5.4. We empirically verify ASIR using a generated dataset of 5,000 cover-stego pairs (with 1,000 pairs reserved for testing). The detection performance is quantitatively evaluated via the minimal average detection error, defined as $\bar{P}_E = \frac{P_{\text{FA}} + P_{\text{MD}}}{2}$, where $P_{\text{FA}}$ and $P_{\text{MD}}$ denote the false alarm and missed detection rates, respectively. Under this metric, a perfectly secure steganographic system forces the detector into random guessing, yielding an ideal $\bar{P}_E$ of 0.50. Against advanced deep learning steganalyzers, ASIR achieves $\bar{P}_E$ scores of 0.5050 (SRNet (Boroumand et al., 2018)), 0.4951 (YeNet (Ye et al., 2017)), and 0.4973 (SiaStegNet (You et al., 2020)). These results closely approach the theoretical optimum. Combined with the min-

*Table 2.* Comprehensive performance evaluation of ASIR across different thresholds($\tau$) and candidate pools($\mathcal{CP}$): capacity(bit), accuracy(Acc), and security ($\bar{P}_E$).

| Datasets | $\tau$ | 0.0 | | | 0.2 | | | 0.4 | | | 0.6 | | |
|---|---|---|---|---|---|---|---|---|---|---|---|---|---|
| | $\mathcal{CP}$ | {2} | {2,3} | {2,3,4} | {2} | {2,3} | {2,3,4} | {2} | {2,3} | {2,3,4} | {2} | {2,3} | {2,3,4} |
| Church | Capacity | 65536 | 103874 | 131072 | 47347 | 65108 | 71059 | 32775 | 38671 | 39255 | 21176 | 22564 | 22592 |
| | Acc(%) | 99.0 | 98.6 | 98.0 | 99.4 | 99.3 | 99.2 | 99.7 | 99.7 | 99.7 | 99.9 | 99.9 | 99.9 |
| | $\bar{P}_E$ | 0.50 | 0.49 | 0.50 | 0.49 | 0.50 | 0.50 | 0.49 | 0.50 | 0.50 | 0.49 | 0.50 | 0.50 |
| Bedroom | Capacity | 65536 | 103874 | 131072 | 32618 | 56807 | 60674 | 25794 | 29411 | 29684 | 14620 | 15338 | 15351 |
| | Acc(%) | 97.0 | 95.6 | 95.1 | 97.7 | 97.3 | 97.0 | 99.0 | 98.9 | 98.9 | 99.7 | 99.7 | 99.6 |
| | $\bar{P}_E$ | 0.50 | 0.49 | 0.50 | 0.49 | 0.50 | 0.49 | 0.51 | 0.49 | 0.51 | 0.50 | 0.49 | 0.50 |
| CelebaHQ | Capacity | 65536 | 103874 | 131072 | 47699 | 65352 | 70481 | 31856 | 36660 | 36998 | 19093 | 20001 | 20013 |
| | Acc(%) | 99.2 | 98.9 | 98.8 | 99.5 | 99.5 | 99.4 | 99.7 | 99.7 | 99.7 | 99.9 | 99.9 | 99.8 |
| | $\bar{P}_E$ | 0.50 | 0.49 | 0.49 | 0.50 | 0.51 | 0.50 | 0.51 | 0.49 | 0.50 | 0.50 | 0.49 | 0.50 |

imal FID scores, this confirms that ASIR is statistically indistinguishable from natural cover images and highly robust against state-of-the-art steganalysis.

## 6.3. Comparison with State-of-the-Art

**Universal Compatibility and Performance Superiority.** To validate ASIR's adaptability across diverse generative architectures, we benchmark it against representative Pixel-Space and Latent-Space methods using parameters $\mathcal{CP} = \{2\}$ and $\tau = 0.0$. As summarized in Table 3, ASIR demonstrates a significant performance leap irrespective of the model type. In Pixel-Space, ASIR outperforms baselines by a substantial margin, offering over an order of magnitude higher capacity compared to Pulsar (Jois et al., 2024) and achieving a superior balance of accuracy and security compared to Diffusion-Stego (Kim et al., 2025). In Latent-Space, ASIR distinctly outperforms Diffusion-Stego by quadrupling the embedding capacity. Furthermore, compared to LDStega (Peng et al., 2024), ASIR maintains the high capacity while significantly improving security to the optimal level ($P_E = 0.50$) and enhancing accuracy, thereby proving its robustness across different diffusion backbones.

**Robustness Evaluation.** We implement ASIR within a latent-based architecture, leveraging the generative decoder to preserve structural semantics while our iterative extraction mechanism corrects potential bit errors. Configured with a stability-prioritized threshold under the candidate pool setting $\mathcal{CP} = \{2\}$ (detailed data are provided in Appendix Table 7), we evaluate robustness against PNG encoding, JPEG compression ($Q = 90$), Resizing (scale 0.5), and Random Dropout (ratio 0.5). As shown in Table 4, ASIR exhibits exceptional resilience, achieving perfect recovery (1.00) for PNG and sustaining 0.99 accuracy under aggressive attacks. Crucially, ASIR maintains this high robustness while offering a significantly larger embedding capacity compared to state-of-the-art methods (Yang et al., 2024b; Xu et al., 2025), demonstrating its superiority in realistic

*Table 3.* Quantitative comparison across Pixel-based and Latent-based diffusion architectures. We evaluate capacity, accuracy(Acc), and security ($\bar{P}_E$) with ASIR settings fixed at $\mathcal{CP} = \{2\}$ and $\tau = 0.0$.

| Model Type | Method | Capacity(bit) | Acc(%) | $\bar{P}_E$ |
|---|---|---|---|---|
| Pixel-based | Pulsar | 3610 | 94.5 | 0.50 |
| | Diffusion-Stego | 65536 | 93.2 | 0.42 |
| | **ASIR** | **65536** | **99.0** | **0.50** |
| Latent-based | LDStega | 16384 | 97.6 | 0.32 |
| | Diffusion-Stego | 4096 | 96.8 | 0.44 |
| | **ASIR** | **16384** | **98.8** | **0.50** |

*Table 4.* Capacity and robustness comparison under common image processing attacks.

| Method | Capacity | PNG | JPEG(90) | Resize(0.5) | Drop(0.5) |
|---|---|---|---|---|---|
| GS | 256 | 1.00 | 1.00 | 1.00 | 1.00 |
| MDDM | 256 | 1.00 | 0.99 | 0.99 | 0.99 |
| **ASIR** | **2070** | 1.00 | 0.99 | 0.99 | 0.99 |

transmission scenarios.

**Computational Efficiency Analysis.** To evaluate ASIR's operational efficiency, we benchmarked its encoding and decoding latency (Figure 5). Under the setting $\mathcal{CP} = \{2\}$ and $\tau = 0.0$, ASIR achieves an encoding time of 1.5s and a decoding time of 3.9s. Its end-to-end latency (5.4s) is comparable to StegaDDPM and yields an approximate 3× extraction speedup over Pulsar (11.8s decoding time), effectively avoiding the latter's computational bottlenecks caused by complex preprocessing heuristics and ECC operations. Despite incorporating gradient-based optimization, ASIR's Iterative Recovery remains remarkably lightweight due to three algorithmic designs: (1) a minimal optimization path restricted exclusively to the final denoising step (from $t = 1$ to $t = 0$); (2) an early stopping strategy that terminates iterations upon loss convergence; and (3) highly parallelized spatial operations that jointly optimize all mes-

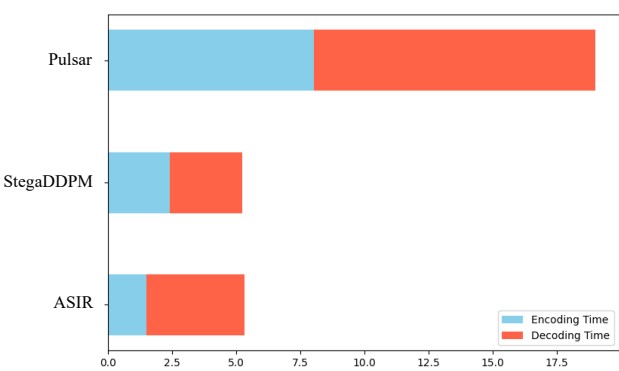

*Figure 5.* Encoding and decoding time comparison for ASIR and baseline methods.

*Table 5.* Ablation study results showing the impact of each component on extraction accuracy.

| Embedding Strategy | Extraction Method | Acc(%) |
|---|---|---|
| Random Sampling | Direct Comparison | 68.7 |
| Antipodal Sampling | Direct Comparison | 76.7 |
| Random Sampling | Iterative Recovery | 94.6 |
| **Antipodal Sampling** | **Iterative Recovery (ASIR)** | **99.0** |

sage bits within a single tensor, rendering extraction time largely independent of payload size. Ultimately, ASIR improves message extraction performance with a reasonable time overhead.

### 6.4. Ablation Study

To validate the effectiveness of our two core contributions, we conduct an ablation study to isolate and quantify the individual impacts of Antipodal Sampling and Iterative Recovery. The experiments were performed on the `CelebaHQ` dataset, with results summarized in Table 5.

**Contribution of Antipodal Sampling** First, we assess the benefit of our structured embedding strategy. As shown in Table 5, when using a simple direct extraction method, switching from a naive random sampling baseline to our Antipodal Sampling improves accuracy from 68.7% to 76.7%. This significant gain confirms that creating a more distinguishable embedding space from the outset is crucial for reducing the initial bit error rate.

**Contribution of Iterative Recovery** Next, we evaluate the impact of our gradient-based extraction method. The results show that Iterative Recovery provides a dramatic improvement in all scenarios. When applied to the naive random sampling method, it boosts accuracy from 68.7% to 94.6%. More importantly, when combined with Antipodal Sampling (the full ASIR model), it elevates the accuracy from 76.7% to a near-perfect 99.0%. This highlights its power in correcting the complex, non-linear distortions from the generation process, which simple comparison methods cannot address.

In addition to the aforementioned analysis of core components, we conduct more extensive ablation studies regarding the compatibility with various sampling schedulers and the strategic selection of embedding timesteps to verify the generalization and optimality of our framework. The detailed

quantitative results and corresponding discussions are deferred to Appendix B.3 and B.4. These studies confirm that the high performance of ASIR is not due to a single component, but to the synergistic combination of both: Antipodal Sampling creates a robust initial signal, and Iterative Recovery refines the extraction to maximize accuracy.

### 7. Conclusion

We propose ASIR to address signal attenuation in diffusion generation by maximizing geometric separability via Antipodal Sampling and reformulating decoding as gradient-based Iterative Recovery. As a training-free, provably secure framework compatible with both pixel and latent models, ASIR establishes a new paradigm simultaneously achieving high capacity, accuracy, and security. As a representative symmetric steganography paradigm, ASIR remains constrained by its dependency on the synchronization of execution environments between the sender and receiver. A detailed analysis and discussion regarding this limitation is provided in Appendix B.5.

### Acknowledgements

This work was supported in part by the National Natural Science Foundation of China under Grant 62302146 and in part by the Fundamental Research Funds for the Central Universities of China under Grant PA2025IISL0104.

The authors would like to thank the anonymous reviewers for their valuable comments and suggestions, which significantly improved the quality of this paper.

### Impact Statement

This work aims to advance secure communication for privacy protection, anti-censorship, and copyright verification. While acknowledging the dual-use risks of steganography, we believe that rigorously exploring its capacity and security limits contributes to a deeper understanding of the field, which is essential for developing future steganalysis and forensic detection methods.

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

## A. Security Analysis

The fundamental requirement of steganographic security is that the stego medium should be statistically indistinguishable from the cover medium. We analyze the security of ASIR based on the complexity-theoretic framework proposed by Hopper et al. (Hopper et al., 2002).

**Problem Definition.** Let $\mathcal{D}_C$ denote the distribution of natural cover images generated by the diffusion model using standard random sampling, and $\mathcal{D}_S$ denote the distribution of stego images generated via ASIR with a secret key $k$. An adversary $\mathcal{A}$, without access to $k$, attempts to distinguish between samples drawn from $\mathcal{D}_C$ and $\mathcal{D}_S$. The steganographic system is considered secure if the adversary's advantage is negligible:

$$|\Pr[\mathcal{A}(x) = 1 \mid x \sim \mathcal{D}_S] - \Pr[\mathcal{A}(x) = 1 \mid x \sim \mathcal{D}_C]| \leq \epsilon \tag{9}$$

where $\epsilon$ is a negligible function of the key length.

**Distributional Equivalence Proof.** In diffusion models, the generation of $x_{t-1}$ from $x_t$ is determined by the predicted mean $\mu_\theta(x_t, t)$ and the variance noise $z$. Since $\mu_\theta$ is deterministic given $x_t$, any statistical discrepancy between cover and stego images must originate from the distribution of the injected noise.

In the standard generation process (Cover), the noise $z$ is sampled directly from the standard normal distribution: $z \sim \mathcal{N}(0, I)$. In the ASIR framework (Stego), the noise $z^{stego}$ is generated via Antipodal Sampling. We prove that $z^{stego}$ follows the exact same distribution $\mathcal{N}(0, I)$:

1. *Uniformity of the Anchor:* The random anchor $r$ is generated by a cryptographically secure pseudo-random number generator (CSPRNG) keyed by $k$. Thus, for an adversary without the key, $r$ is computationally indistinguishable from a uniform variable $U \sim \mathcal{U}[0, 1)$.

2. *Preservation of Uniformity under Shift:* The message embedding modifies the probability value to $p(m) = (r + \delta_m)$ (mod 1), where $\delta_m = m/R$ is the message-dependent shift. A fundamental property of the uniform distribution is that adding a constant (modulo 1) to a uniform variable preserves the uniform distribution. Therefore, the marginal distribution of $p(m)$ remains $\mathcal{U}[0, 1)$, regardless of the message bit $m$.

3. *Inverse Probability Transform:* ASIR applies the Inverse Cumulative Distribution Function (ICDF) of the standard Gaussian, denoted as $\Phi^{-1}$, to obtain the noise: $z^{stego} = \Phi^{-1}(p(m))$. By the Probability Integral Transform theorem, applying $\Phi^{-1}$ to a uniformly distributed variable produces a variable that follows the standard Gaussian distribution exactly.

**Conclusion.** Since the marginal distribution of the noise $z^{stego}$ is mathematically identical to the natural noise $z \sim \mathcal{N}(0, I)$, the resulting transition probability $P(x_{t-1}|x_t)$ in ASIR is identical to that of the standard diffusion process. Consequently, the generated stego images are statistically indistinguishable from cover images. The security of ASIR effectively reduces to the security of the underlying CSPRNG. Without the secret key $k$, an adversary cannot distinguish the structural relationship between antipodal points from pure randomness, ensuring computational security.

## B. Experiments

### B.1. Quantitative Image Quality Analysis

To rigorously evaluate the impact of the ASIR embedding process on visual fidelity, we conducted a quantitative analysis using the Fréchet Inception Distance (FID). We compared the distributional distance of generated images against their respective training datasets for both the baseline generation (Cover) and the ASIR-embedded generation (ASIR).

Table 6 presents the detailed FID scores across four different model architectures and datasets. The "Cover" column represents the FID of images generated by standard sampling, while the "ASIR" column represents the FID of stego images containing embedded messages.

As observed, the increase in FID scores introduced by ASIR is marginal across all tested scenarios. For instance, on the Stable Diffusion-v2.1 model, the difference is merely $\sim 0.2$, and even on pixel-space DDPMs, the deviation remains minimal. This quantitative evidence supports our qualitative findings, confirming that ASIR preserves the statistical properties of the generated distribution and maintains high perceptual quality.

*Table 6.* Quantitative comparison of FID scores between standard generation (Cover) and ASIR-based steganography (ASIR). Lower FID indicates better image quality and higher similarity to the real data distribution.

| Model | Dataset | Cover | ASIR |
|---|---|---|---|
| Stable Diffusion-v2.1 | COCO | 28.1647 | 28.3605 |
| DDPM-Church | LSUN(Church) | 15.9739 | 17.8049 |
| DDPM-Bedroom | LSUN(Bedroom) | 20.8273 | 21.8627 |
| DDPM-CelebA-HQ | CelebA-HQ | 44.4197 | 45.8253 |

## B.2. Embedding Capacity and Extraction Accuracy Analysis

In the main text, we adopted a redundancy setting of $R = 2$, which corresponds to a candidate pool size of $\mathcal{CP} = \{2\}$. To provide a comprehensive evaluation of ASIR's behavior under this configuration, we conducted supplementary experiments varying the truncation threshold $\tau$. Table 7 details the Capacity, Accuracy, and Security ($\bar{P}_E$) metrics across both Pixel-based and Latent-based architectures.

For DDPMs trained on Church, Bedroom, and CelebA-HQ, the results indicate a stable performance progression. At $\tau = 0.0$, the method utilizes the full embedding potential, achieving the theoretical maximum capacity of 65,536 bits with high accuracy. Increasing $\tau$ serves to progressively refine the selection of message bits based on confidence. Notably, at $\tau = 0.6$, ASIR consistently achieves near-perfect recovery (Acc $\approx 99.9\%$) while retaining a substantial capacity range of 14k–21k bits.

Experiments on Stable Diffusion (v1.5 and v2.1) reveal that Latent-based architectures are highly responsive to threshold adjustments. A minimal increase in $\tau$ allows the system to transition from a high-capacity state (16,384 bits at $\tau = 0.00$) to a zero-error state. Specifically, with a slight adjustment to $\tau \in [0.04, 0.06]$, the extraction accuracy reaches $100.0\%$. Throughout all configurations, the security metric $\bar{P}_E$ remains consistently close to the ideal value of 0.50, verifying that parameter tuning does not compromise the statistical imperceptibility of the steganography.

*Table 7.* Impact of truncation threshold $\tau$ on Capacity, Accuracy, and Security ($\bar{P}_E$) across Pixel-based and Latent-based architectures with fixed $\mathcal{CP} = \{2\}$.

| Architecture | Model | $\tau$ | Capacity (bits) | Acc (%) | $\bar{P}_E$ |
|---|---|---|---|---|---|
| Pixel-based | DDPM-Church | 0.0 | 65536 | 99.0 | 0.50 |
| | | 0.2 | 47347 | 99.4 | 0.49 |
| | | 0.4 | 32775 | 99.7 | 0.50 |
| | | 0.6 | 21176 | 99.9 | 0.50 |
| | DDPM-Bedroom | 0.0 | 65536 | 97.0 | 0.50 |
| | | 0.2 | 32618 | 97.7 | 0.50 |
| | | 0.4 | 25794 | 99.0 | 0.51 |
| | | 0.6 | 14620 | 99.7 | 0.50 |
| | DDPM-CelebA-HQ | 0.0 | 65536 | 99.2 | 0.49 |
| | | 0.2 | 47699 | 99.5 | 0.50 |
| | | 0.4 | 31856 | 99.7 | 0.50 |
| | | 0.6 | 19093 | 99.9 | 0.50 |
| Latent-based | Stable Diffusion-v2.1 | 0.00 | 16384 | 98.8 | 0.50 |
| | | 0.04 | 2070 | 100.0 | 0.49 |
| | | 0.06 | 121 | 100.0 | 0.50 |
| | Stable Diffusion-v1.5 | 0.00 | 16384 | 98.6 | 0.50 |
| | | 0.04 | 2076 | 99.9 | 0.50 |
| | | 0.06 | 116 | 100.0 | 0.51 |

## B.3. Compatibility with Sampling Schedulers

In the primary experiments of this paper, we default to using the DDIM scheduler for both image generation and message extraction. To further validate the generality of the ASIR framework, we extend our analysis to evaluate its compatibility with different sampling schedulers.

From a theoretical perspective, ASIR can be directly applied as long as the sampling process of the diffusion model adheres to the decomposed structure of "deterministic prediction $\mu_\theta$ combined with random variance noise $z$". Specifically, ASIR is compatible with all schedulers that incorporate a random noise injection mechanism, including DDPM, DDIM (when $\sigma_t > 0$), and DPM++ SDE. It is important to note that for purely deterministic ODE solvers, variance noise is not introduced into their sampling trajectories, resulting in a lack of carriers for masking and bearing messages. Consequently, ASIR cannot be directly applied to such deterministic samplers.

We evaluated the extraction accuracy of ASIR under various scheduler configurations. The validation results, as presented in Table 8, demonstrate that ASIR maintains a high degree of stability across the DDPM, DDIM, and DPM++ SDE schedulers. This proves that the proposed framework is not confined to a single scheduler and exhibits robust generalization capabilities.

*Table 8.* Extraction accuracy (Acc) comparison of ASIR across different stochastic sampling schedulers.

| Datasets | DDIM | DDPM | DPM++ SDE |
|---|---|---|---|
| Church | 99.0% | 99.0% | 99.0% |
| Bedroom | 97.0% | 97.6% | 97.6% |
| CelebA-HQ | 99.2% | 99.1% | 99.1% |

## B.4. Ablation on Embedding Timesteps

In the core framework described in Section 5, we establish the message embedding node at the second-to-last denoising timestep of the diffusion process (i.e., $t = 1$). To demonstrate the optimality of this configuration, we conducted ablation studies focusing on the embedding timestep. Utilizing the CelebA-HQ dataset, we evaluated the performance when the message injection point is shifted to earlier timesteps.

*Table 9.* Ablation study results on message extraction accuracy across different embedding timesteps.

| Time Step | Embedding and Extraction Strategy | | | |
|---|---|---|---|---|
| | Random Sampling Direct Comparison | Antipodal Sampling Direct Comparison | Random Sampling Iterative Recovery | Antipodal Sampling Iterative Recovery (ASIR) |
| the fifth-to-last step | 52.2% | 53.7% | 71.2% | **74.6%** |
| the fourth-to-last step | 52.5% | 54.0% | 77.6% | **81.8%** |
| the third-to-last step | 54.8% | 59.5% | 86.1% | **92.5%** |
| the second-to-last step | 68.7% | 76.7% | 94.6% | **99.0%** |

**Experimental Conclusions and Analysis:**

The test results clearly indicate that embedding at the second-to-last step maximizes the performance advantages of ASIR, achieving an accuracy of 99.0%. This specific time point maximally circumvents the severe signal washout occurring in early denoising steps and minimizes the distortion introduced by the generation pipeline, thereby serving as the optimal system setting.

More importantly, this ablation study highlights the exceptional robustness of ASIR under extreme channel attenuation. When the embedding point is advanced to earlier timesteps, the signal is subjected to severe washout across multiple denoising iterations. Under these conditions, the baseline "Random Sampling + Direct Comparison" strategy rapidly degrades, with its accuracy plummeting towards the 50% random guessing threshold. Nevertheless, even in the highly attenuating environment of the fifth-to-last step, ASIR leverages the maximized initial signal separation provided by Antipodal Sampling and the robust gradient correction of Iterative Recovery to substantially elevate and sustain the accuracy at 74.6% (and even reaches 92.5% at the third-to-last step).

This stark contrast not only validates the rationality of the default embedding point but also fundamentally proves that the performance leap achieved by ASIR within complex generative models is not a mere opportunistic circumvention of the denoising steps. Rather, it underscores that the core mechanism itself possesses remarkably strong anti-distortion and signal recovery capabilities.

### B.5. Discussion and Future Work

**Synchronization Dependency** Consistent with other generative steganography frameworks (Jois et al., 2024; Peng et al., 2023), ASIR follows the symmetric steganographic paradigm. The primary design objective is to achieve maximum embedding capacity and extraction accuracy. Mechanistically, such methods rely on the precise reconstruction of diffusion trajectories, which necessitates perfectly synchronized model weights and scheduler configurations between the sender and receiver. Consequently, this strict requirement for environmental synchronization, and the resulting limitation on robustness under unknown generation pipelines, represents the inherent technical bottleneck of the symmetric paradigm.

Regarding the practical feasibility of maintaining synchronization across diverse platforms, our experimental results are presented in Table 10. The evaluation demonstrates that provided the model parameters and software execution environments remain consistent, ASIR's extraction accuracy remains robust even when the sender and receiver utilize different hardware architectures (GPU vs. CPU) or calculation precisions (FP32 vs. FP16). This indicates that physical hardware variations do not lead to trajectory divergence when the computational environment is controlled. In practical deployment, containerization technologies (e.g., Docker) provide a mature solution for ensuring such environmental consistency. By fully encapsulating all software dependencies, these tools ensure that generative processes are highly reproducible.

*Table 10.* Extraction accuracy (Acc) of ASIR across different hardware architectures and computational precisions under a unified software environment.

| Sender | Receiver | Church | CelebA-HQ |
| --- | --- | --- | --- |
| GPU, FP32 | GPU, FP32 | 99.0% | 99.1% |
| **GPU**, FP32 | **CPU**, FP32 | 99.0% | 99.1% |
| GPU, **FP16** | GPU, **FP32** | 99.0% | 99.1% |
| GPU, **FP32** | GPU, **FP16** | 99.0% | 99.1% |

**Towards Asymmetric and Environment-Agnostic Steganography.** While current engineering practices effectively mitigate synchronization issues, the reliance on strict environmental consistency remains a major constraint for open-world deployment. Future research will shift from engineering remediation toward algorithmic breakthroughs. A pivotal direction is the development of Asymmetric Generative Steganography. Unlike the current paradigm where the receiver must mirror the sender's generative parameters, an asymmetric framework aims to decouple the extraction process from the generative backbone. The ultimate goal is to allow the sender to utilize complex generative priors for secure embedding, while the receiver recovers messages through a lightweight, model-agnostic decoder or a universal key, thereby eliminating the dependency on specific model configurations and computational environments.

