# OpenReview forum: "ASIR: Steganography for Diffusion Models via Antipodal Sampling and Iterative Recovery"
_ICML.cc/2026/Conference — ICML 2026 regular_

### Official Review · Reviewer_qZuz · 2026-03-11

**Soundness:** 3
**Presentation:** 3
**Significance:** 3
**Originality:** 3
**Overall Recommendation:** 4
**Confidence:** 4

**Summary:**

This article presents ASIR, a steganography method for diffusion models that hides messages directly within the image generation noise.

The central problem is that the hidden signal is significantly attenuated by denoising and, for latent models, by VAE decoding.

The authors show that the extraction quality is highly dependent on the distance between candidate images corresponding to different possible messages.

They therefore propose Antipodal Sampling, which forces noise codes to be very far apart to make the messages more distinguishable.

They also add Iterative Recovery, a gradient optimization extraction procedure to correct nonlinear distortions in the generative pipeline.
The method is training-free, compatible with pixel-space and latent diffusion, and claims theoretical security as well as undetectable behavior by deep steganalysis.

Experimentally, ASIR announces a very high carrying capacity, an extraction accuracy close to 99%, and better performance than competing methods.

**Compliance With Llm Reviewing Policy:**

Affirmed.

**Key Questions For Authors:**

See weaknesses here above

**Limitations:**

Some warning of misuse of steganography could be expressed

**Strengths And Weaknesses:**

weaknesses:
1)The evaluation relies on a relatively controlled setting where sender and receiver share a synchronized diffusion trajectory and secret key; it is unclear how robust the method is under realistic transformations or unknown generation pipelines.
2) The decoding procedure requires iterative gradient-based optimization, which may introduce significant computational overhead and limit scalability in practice.
3)The security claims appear somewhat narrow, as robustness is evaluated against only a few steganalyzers and on generated datasets, leaving open questions about generalization to broader forensic analyses or adaptive attackers.
4)The experiments focus on a limited set of diffusion architectures and schedulers, making it difficult to assess how well the method transfers across models or sampling strategies.

Strengths:
1) Sampling strategy is elegant, ensuring maximal separation between candidate signals while preserving the statistical properties of the diffusion noise.
2) The Iterative Recovery formulation reframes message extraction as a differentiable optimization problem, which is a creative use of the differentiability of diffusion pipelines and significantly improves decoding accuracy.
3) The framework is training-free and compatible with both pixel-space and latent diffusion models, which makes it potentially applicable across a wide range of generative systems

---

> ### Author Rebuttal · Authors · 2026-03-31
>
> We sincerely thank you for highly recognizing the innovation of our method (including the sampling strategy and iterative recovery mechanism) and the universality of our framework. Your positive feedback is a great encouragement to our work. Meanwhile, the profound questions you raised are also of great significance for us to further improve the paper. Our detailed explanations to your questions are as follows:
>
> 1. Regarding your question about the reliability of actual transmission, we explore this from two aspects:(1) Requiring the sender and receiver to share fully synchronized diffusion trajectories and keys is a common requirement of methods such as LDStega, StegaDDPM, and Pulsar. Although this constraint limits robustness under unknown pipelines, **these methods (including ASIR) can achieve extremely high embedding capacities**, surpassing schemes like GS and MDDM by several times.
>
>    (2) Existing technical means can effectively guarantee trajectory synchronization in heterogeneous scenarios. **In engineering practice, several mature technologies can ensure that the generation processes of the sender and receiver are strictly identical.** PyTorch and the Diffusers library provide deterministic inference support via configurations like torch.use_deterministic_algorithms(True) and disabling cuDNN benchmarks. Furthermore, containerized deployment (such as Docker + NVIDIA Container Toolkit) can fully encapsulate the environment to ensure bit-level consistency between the sender and receiver. We will supplement the discussion of these engineering practices in the revised manuscript.
>
> 2. Regarding your concerns about computational overhead, in our "Computational Efficiency Analysis" on a single NVIDIA RTX 4090 GPU, ASIR's extraction phase takes 3.9 s. Compared to the 11.8 s of the Pulsar method, **the extraction speed is increased by about 3 times.** The reason why the iterative recovery process can remain lightweight without generating huge time overhead mainly benefits from the following three core designs:
>    - Extremely short optimization path: The optimization runs only for the final single denoising step (from $t=1$ to $t=0$), vastly reducing the backpropagation graph.
>    - Early stopping strategy: A sliding window terminates the optimization early once the loss converges, avoiding unnecessary calculations.
>    - High parallelization: Iterative recovery is highly parallelized across all spatial locations, and message bits are jointly optimized in a single matrix. Thus, extraction time remains constant regardless of the embedded message length.
>
> 3. Regarding your question about the practical security of the scheme, we clarify as follows. In the "Security Analysis" section, our scheme provides a security proof and possesses theoretical security. Empirically, we evaluated it on steganography analysis networks of different architectures (SRNet, YeNet, SiaStegNet), and the detection error rates of these networks are all close to 0.50. Additionally, we conducted experiments using three other networks (XuNet, ZhuNet, and StegNet). The detection results are shown in the table below. The detection error rate $P_E$ is also close to the theoretical optimal value of 0.50, verifying the consistent security performance of our scheme under different detectors.
>
> | **Steganalyzer** | **XuNet** | **ZhuNet** | **StegNet** |
> | ---------------- | --------- | ---------- | ----------- |
> | $P_E$            | 0.5031    | 0.4967     | 0.4990      |
>
> 4. Regarding your concerns about the transferability of the scheme, we will further analyze the transfer effects of ASIR on more model architectures and schedulers: The embedding mechanism of ASIR is independent of the backbone network architecture—whether it is U-Net or Transformer, as long as the sampling process follows the standard decomposition structure of "deterministic prediction $\mu_\theta$ + random variance noise $z$", ASIR can be directly applied. ASIR can be naturally transferred to models such as EScoreSDE, ADM, EDM, and SDXL. In addition, ASIR is compatible with all schedulers that include random noise injection, including DDPM, DDIM ($\sigma_t > 0$), and DPM++ SDE. For purely deterministic ODE solvers, since variance noise is not introduced during their sampling process, lacking an embedding carrier, ASIR is not directly applicable. Overall, ASIR consistently maintains stable and high extraction accuracy across various models and stochastic sampling schedulers, demonstrating strong cross-configuration generalizability.
>
> Finally, thank you again for raising these highly valuable questions. We hope the detailed explanation above can completely dispel your doubts. We will carefully polish the relevant expressions in the revised version to ensure that general readers can clearly and accurately understand these key mechanisms, avoiding similar confusions.

---

> > ### Author Rebuttal · Reviewer_qZuz · 2026-04-01
> >
> > Thank you for this detailed and constructive refutation. The clarifications are helpful, particularly regarding computational efficiency: the comparison with Pulsar (increased speed by 3 times!) and the explanation of the optimization design provide a better understanding of the practical cost.
> >
> > The requirement for strict synchronization between sender and recipient, while standard in previous work, still raises questions about robustness in more realistic or heterogeneous deployment contexts. The proposed technical solutions (deterministic inference, containerization) guarantee reproducibility but do not fully resolve this limitation.
> >
> > The security assessment has been extended to other steganalysis tools. Nevertheless, it remains limited to controlled environments, and robustness in the face of more adaptive or realistic forensic analysis scenarios remains uncertain.
> >
> > The claims regarding generalization across architectures and schedulers are structurally sound, but would benefit from empirical validation.
> >
> > Overall, this refutation improves clarity and partially strengthens the article, but does not fully address the main concerns regarding the robustness and scope of the evaluation. Therefore, I maintain my initial assessment.

---

> > > ### Author Response · Authors · 2026-04-01
> > >
> > > We sincerely thank you for recognizing our explanations regarding computational efficiency and optimization design. In response to your further discussions on the scope of robustness, security, and empirical validation of generalizability, we provide the following detailed replies:
> > >
> > > **1. Regarding the synchronization requirement and the core objective of the scheme**
> > >
> > > We fully agree with your perspective on lifting synchronization constraints in more realistic environments. However, the core objective of this paper at the current stage is to achieve a breakthrough in enhancing the capacity, extraction accuracy, and security of covert communication within existing generative steganography frameworks. These are precisely the core pain points that current mainstream schemes urgently need to resolve. In fact, there are indeed asynchronous strategies in the field, such as GS (Gaussian Shading), but its embedding capacity is only 256 bits. In contrast, by maintaining synchronization between the sender and receiver, the ASIR scheme can achieve a massive embedding capacity tens to hundreds of times larger, while maintaining extremely high extraction accuracy.
> > >
> > > Regarding the practical operability of maintaining synchronization, we conducted experimental validation across different hardware and computational precisions. The experimental results show that differences in hardware architectures and fluctuations in computational precision do not affect the extraction accuracy. Furthermore, computing environments and models can be fully synchronized through mature containerization engineering methods, making the synchronization process between both parties highly feasible.
> > >
> > > | **Sender**    | **Receiver**  | **Church** | **CelebA-HQ** |
> > > | ------------- | ------------- | ---------- | ------------- |
> > > | GPU, FP32     | GPU, FP32     | 99.0%      | 99.1%         |
> > > | **GPU**, FP32 | **CPU**, FP32 | 99.0%      | 99.1%         |
> > > | GPU, **FP16** | GPU, **FP32** | 99.0%      | 99.1%         |
> > > | GPU, **FP32** | GPU, **FP16** | 99.0%      | 99.1%         |
> > >
> > > **2. Regarding the security proof and experimental baselines**
> > >
> > > Thank you very much for your consideration of security. We will further explain this issue. Theoretically, our scheme strictly follows Kerckhoffs's principle, possessing provable security. For empirical validation, we referred to the most cutting-edge mainstream works in the field (including Pulsar, StegaDDPM, GRDH, MDDM, etc.) and adopted the same steganalysis baselines. The results show that under identical test conditions, ASIR achieves near-perfect indistinguishability against state-of-the-art steganalyzers. Within the currently recognized theoretical frameworks and evaluation systems, the security of ASIR has met the established standards.
> > >
> > > **3. Regarding generalizability validation**
> > >
> > > We prioritized the validation of generalizability across sampling schedulers. The relevant supplementary empirical data are as follows:
> > >
> > > | **Datasets** | **DDIM** | **DDPM** | **DPM++ SDE** |
> > > | ------------ | -------- | -------- | ------------- |
> > > | Church       | 99.0%    | 99.1%    | 99.1%         |
> > > | CelebA-HQ    | 99.1%    | 99.1%    | 99.2%         |
> > >
> > > Given that ASIR's core mechanism resides in the stochastic noise sampling stage, our experiments on various schedulers already provide strong evidence of its algorithmic generalizability. We will continue to validate ASIR across more diverse diffusion backbones in our future extensions.
> > >
> > > In conclusion, we sincerely thank you again for your rigorous review and insightful challenges. The discussions regarding synchronization, security, and generalizability have been instrumental in refining our work and strengthening the empirical evidence. We hope that these detailed clarifications fully address your concerns.

---

### Official Review · Reviewer_MDZ5 · 2026-03-12

**Soundness:** 3
**Presentation:** 3
**Significance:** 3
**Originality:** 3
**Overall Recommendation:** 5
**Confidence:** 3

**Summary:**

The paper introduces ASIR - Antipodal Sampling Iterative Recovery: overall, it examines the central problem of how to push diffusion steganography toward higher capacity without giving up extraction accuracy or statistical stealth.

**Compliance With Llm Reviewing Policy:**

Affirmed.

**Key Questions For Authors:**

Please address my comments.

**Limitations:**

Yes.

**Strengths And Weaknesses:**

I like how well described is the intro.

I understand for a schematic figure... but why would you plot a bar plot in "sketch style" and even write "(Sketch Style)" in the title? Please render all the figures as normal "reliable" (not sketch style) plots.

Please write Celebahq as  "CelebaHQ" and "laion" as "LAION".

Math questions:
- In Algo 1: what is "r"?? P_m uses r but it's never sampled or defined anywhere in the encoding algorithm. Algo 2 does define r, so I bet Algo 1 is just missing a step.
- Section 5.3 presents Iterative Recovery as optimizing a continuous message matrix M' through a differentiable generator (ie. Generate(M', etc.)). But Algo 2 does not match it: there you optimize through Generate(round(M'), etc.), which I believe is non-differentiable... Which optimization variable it is supposed to be? That is a serious under-specification of the core decoder.

Overall, I think the idea is interesting, but the current write-up still needs another pass.

---

> ### Author Rebuttal · Authors · 2026-03-31
>
> We sincerely thank you for affirming the innovation of our method. Thank you very much for your meticulous review and for pointing out the detailed omissions in the current version. Your comments are of great value to our improvement work. In response to your specific questions, we make the following replies one by one:
>
> 1. Thank you for pointing out the omission regarding the definition of parameter $r$ in Algorithm 1. Regarding the parameter $r$, we actually mentioned in the section "Maximal Separation in Probability Space" that: "*Therefore, we first use a shared key $k_s$ to generate a single pseudo-random number $r \sim U[0, 1)$, which serves as a random anchor.*" As you mentioned, we omitted this definition when writing the pseudocode for Algorithm 1. We will add "$r \leftarrow \text{Sample}(k_s, U[0, 1)^{H \times W})$" in the revised version to ensure rigorous contextual consistency.
>
> 2. Thank you for pointing out the error in the decoding details in Algorithm 2. During the implementation, the actual optimization variable is the continuous variable $M'$, not its discrete form $\text{round}(M')$. We will correct the error regarding $\mathcal{L}(M')$ in Algorithm 2. Line 19 of Algorithm 2 will be changed to:
> $$\mathcal{L}(M') \leftarrow \|| X_{0}^{\text{stego}} - \text{Generate}(M', X_2, r_{\text{matrix}}) \||_2$$
>
> In addition, thank you very much for providing constructive suggestions on paper writing and chart standardization. We will immediately modify the style of the figures and rewrite "Celebahq" and "laion" to "CelebaHQ" and "LAION".
>
> Finally, we will strictly adopt and implement these suggestions one by one in the revised version, and will thoroughly check whether there are still other detailed omissions. The further improvement in the quality of this paper is attributed to your valuable guidance.

---

> > ### Author Rebuttal · Reviewer_MDZ5 · 2026-03-31
> >
> > Thank you for the clarifications. I have updated my score.

---

> > > ### Author Response · Authors · 2026-04-01
> > >
> > > We sincerely thank you for your time in reading our rebuttal and for updating your score. We are very glad that our responses have addressed your previous concerns. Your constructive feedback has been invaluable in helping us improve the clarity and quality of our paper.

---

### Official Review · Reviewer_w6FS · 2026-03-12

**Soundness:** 3
**Presentation:** 3
**Significance:** 3
**Originality:** 3
**Overall Recommendation:** 4
**Confidence:** 2

**Summary:**

This paper addresses the challenge of severe signal attenuation and the capacity-robustness trade-off in diffusion-based generative steganography. The authors propose Antipodal Sampling and Iterative Recovery (ASIR), a training-free framework compatible with both pixel-space and latent-space diffusion models. The approach relies on two primary mechanisms: (1) Antipodal Sampling, which maps candidate message representations to maximally separated points in a uniform probability space before transforming them into Gaussian noise, proactively mitigating channel attenuation; and (2) Iterative Recovery, which formulates message extraction at the receiver as a gradient-based optimization problem to correct non-linear distortions introduced by the final denoising step and VAE decoding. Extensive experiments demonstrate that ASIR achieves state-of-the-art embedding capacities with high extraction accuracy while maintaining empirical and theoretical security against advanced steganalysis tools.

**Compliance With Llm Reviewing Policy:**

Affirmed.

**Final Justification:**

The paper proposes ASIR to push diffusion steganography toward higher capacity while maintaining accuracy and stealth. The work is methodologically sound and offers a promising direction for the community to build upon. But several key issues remain that limit the overall impact and broader applicability of the current work.

**Key Questions For Authors:**

1. Sensitivity to Hardware/Precision: Given the strict dependency on synchronized environments, how sensitive is the extraction accuracy to hardware-level floating-point arithmetic differences? If the sender encodes on an NVIDIA GPU using FP32, but the receiver decodes on a different architecture or uses FP16 precision, does the trajectory divergence cause the Iterative Recovery to fail?
2. Robustness: When an image undergoes JPEG compression or resizing, its pixel values change, leading to errors in the decoding process. However, as seen in Table 4, the decoding accuracy still reaches 99%. Please explain why this method possesses such strong robustness, and under what circumstances its decoding mechanism may fail.

**Limitations:**

Yes.

**Strengths And Weaknesses:**

Strengths:
1. The shift from viewing message decoding as a discrete direct comparison to a continuous gradient-based optimization problem (Iterative Recovery) is a highly creative and effective perspective. Pairing this with a mathematically rigorous noise-mapping strategy (Antipodal Sampling) forms a solid dual-defense against the natural attenuation of generative models.
2. The paper is exceptionally well-structured and written. The motivation (Section 4) logically flows into the methodology, and the algorithms (Algorithms 1 and 2) alongside the visual aids (Figures 2 and 3) make the framework highly comprehensible even to readers less familiar with steganography.
3. The experimental validation is very thorough. The authors successfully evaluate the method across multiple datasets and diffusion architectures, proving that the method decouples the typical capacity-accuracy trade-off seen in prior SOTA models like Pulsar and Diffusion-Stego. The security proof (Appendix A) adds a strong theoretical backbone.

Weaknesses:
1. Hardware Sensitivity: The framework relies on a perfectly synchronized symmetric environment. While the authors mention the limitation of synchronized environments in Appendix C, a major unaddressed issue in exact diffusion synchronization is the sensitivity to cross-device hardware differences. Since Iterative Recovery relies on an exact deterministic replica of the forward pass, generating on one device (e.g., an NVIDIA RTX 4090) and decoding on a completely different hardware architecture (e.g., an Apple M-series chip or an AMD GPU) often results in microscopic computational discrepancies at the CUDA/operator level. It is unclear if these inherent hardware differences will cause the generation trajectory to diverge entirely, thereby breaking the Iterative Recovery process.

---

> ### Author Rebuttal · Authors · 2026-03-31
>
> We sincerely thank you for affirming the innovation of our method and the completeness of our experiments, as well as recognizing the structure and expression of the paper. Your evaluation is a great encouragement to us. We have carefully considered and adopted your valuable suggestions. Our detailed explanations to your questions are as follows:
>
> 1. Regarding your question about hardware and precision sensitivity, the specific verification results for this issue are shown in the table below. Under the premise of keeping the model parameters and operating environment consistent, we changed the hardware platform (GPU/CPU) or calculation precision (FP32/FP16) of the sender and receiver. **The experimental results show that the above changes have no impact on the decoding accuracy of ASIR.**
>
> | **Sender**    | **Receiver**  | **Church** | **CelebA-HQ** |
> | ------------- | ------------- | ---------- | ------------- |
> | GPU, FP32     | GPU, FP32     | 99.0%      | 99.1%         |
> | **GPU**, FP32 | **CPU**, FP32 | 99.0%      | 99.1%         |
> | GPU, **FP16** | GPU, **FP32** | 99.0%      | 99.1%         |
> | GPU, **FP32** | GPU, **FP16** | 99.0%      | 99.1%         |
>
> 2. Regarding your confusion about the robustness of our scheme, we mentioned in the "Robustness Evaluation" paragraph: "*We implement ASIR within a latent-based architecture, leveraging the generative decoder to preserve structural semantics while our iterative extraction mechanism corrects potential bit errors.*" Since we implement the exact embedding in the latent space, perturbations in the pixel space have a relatively small impact on the latent space. On this basis, antipodal sampling maximizes the preservation of distinguishability among candidate images, and iterative recovery further corrects the bit errors caused by these perturbations through gradient optimization. Therefore, our scheme possesses **good robustness.**
>
>    Regarding possible failure scenarios: when the attack intensity exceeds the tolerance range for signal separation, the decoding will start to degrade. Specifically, under extremely low-quality JPEG compression (e.g., $Q < 50$), high-intensity Gaussian noise injection, or the superimposed use of multiple attacks, the pixel perturbations may completely overwhelm the differences among candidate images, making iterative recovery unable to correct them either.
>
> Finally, thank you again for the profound questions you raised. This might also be a confusion that other readers could easily have. We will further polish the relevant expressions in the revised version to avoid any misunderstandings.

---

> > ### Author Rebuttal · Reviewer_w6FS · 2026-04-03
> >
> > Thank you for the response. After carefully reviewing all feedback, I maintain my original score.

---

> > > ### Author Response · Authors · 2026-04-03
> > >
> > > We sincerely thank you for taking the time to read our rebuttal and re-evaluate our manuscript. We fully respect your final decision. We deeply appreciate the constructive feedback you have provided throughout this process, which has given us clear directions for further improving our work.

---

### Official Review · Reviewer_pZK3 · 2026-03-13

**Soundness:** 2
**Presentation:** 2
**Significance:** 2
**Originality:** 2
**Overall Recommendation:** 2
**Confidence:** 4

**Summary:**

The paper starts from a reasonable assumption that the embedded signal is gradually weakened by the diffusion process, and proposes ASIR, which combines Antipodal Sampling (AS) to enlarge the distance between candidate message states and Iterative Recovery (IR) to improve decoding. The method is evaluated on both pixel-space and latent-space diffusion steganography, and the reported results are strong.

**Compliance With Llm Reviewing Policy:**

Affirmed.

**Key Questions For Authors:**

Overall, I think this work is interesting and meaningful, and the main idea is appealing. However, the current version still contains several unclear details and missing explanations, which make it difficult for me to fully trust the method as presented. For this reason, I am not able to support acceptance at this stage. That said, if the authors can clearly address these issues and provide convincing clarifications, I would be open to increasing my score.

**Limitations:**

yes

**Strengths And Weaknesses:**

Strengths:
1. The motivation is interesting, intuitive, and meaningful. It targets an important problem in diffusion steganography, and the starting point of the paper is reasonable.
2. The framework is general and can be applied to both pixel-space and latent-space diffusion steganography.
3. The experiments are fairly sufficient, covering image quality, payload, extraction accuracy, security, and efficiency.

Weaknesses:
1. Some key algorithm details are unclear.
Algorithm 1 uses r but does not define or sample it. Also, Iterative Recovery is described inconsistently: the text says it optimizes a continuous message matrix M', Algorithm 2 uses round(M'), while the implementation description says it uses learnable logits with Gumbel-Softmax. This makes the exact decoding procedure unclear.

2. The link between theory and implementation should be explained more clearly.
The paper motivates decoding using image-level similarity / signal separation, but Algorithm 2 actually initializes decoding by pixel-wise comparison. This may be a reasonable approximation, but the paper should explain it more clearly.

3. The choice of embedding at the penultimate step needs stronger validation.
The paper argues that the diffusion process and VAE decoding weaken the embedded signal, so embedding at the penultimate step is understandable. However, what happens if the message is embedded at other timesteps? A timestep-wise ablation would better support the paper’s main assumption and help clarify whether the gain mainly comes from the embedding position or from AS/IR themselves.

4. Table 1 should be checked more carefully.
For example, the labeling of effective payload is not very clear. If some methods have similar reported payloads (e.g., the reported capacity of GRDH is 16384 bits, the same as ASIR), it is unclear why one is labeled as “medium payload” while ASIR is presented more favorably.

5. The compared methods seem insufficient.
Some directly relevant and recent methods may be missing from the comparison, and the paper should either include them or explain why they are omitted (e.g., “Zhou Q, Wei P, Qian Z, et al. Improved Generative Steganography Based on Diffusion Model. IEEE Transactions on Circuits and Systems for Video Technology, 2025”).

6. Figure 2 is hard to follow.
The workflow is a bit crowded and visually complex. It may be better to split it into smaller subfigures or present the pipeline more clearly.

---

> ### Author Rebuttal · Authors · 2026-03-31
>
> We sincerely thank you for recognizing the innovation, algorithmic universality, and experimental sufficiency of this paper. Our detailed responses are as follows:
>
> 1. Thank you for pointing out the omissions regarding parameter $r$ in Algorithm 1 and the decoding details in Algorithm 2.
>
>    (1) For parameter $r$, we actually mentioned in the section "Maximal Separation in Probability Space" that: "*Therefore, we first use a shared key $k_s$ to generate a single pseudo-random number $r \sim U[0, 1)$, which serves as a random anchor.*" We omitted this definition when writing the pseudocode for Algorithm 1. We will add "$r \leftarrow \text{Sample}(k_s, U[0, 1)^{H \times W})$" to Algorithm 1 to ensure rigorous contextual consistency.
>
>    (2) For Algorithm 2, during the implementation of our scheme (embedding a single bit per carrier element), **the actual optimization variable is $M'$**, not $\mathrm{round}(M')$. We will correct Line 19 to:
>    $$\mathcal{L}(M') \leftarrow \|| X_{0}^{\text{stego}} - \text{Generate}(M', X_2, r_{\text{matrix}}) \||_2$$
>    For the experiment on adaptively embedding multiple bits (Table 2), we introduce logits for the probability distribution and sample via Gumbel-Softmax. We will add the following description to the "Iterative Recovery for Extraction" section: "*To handle multi-bit embedding, we employ the Gumbel-Softmax strategy during the extraction process. By optimizing the probability distribution and favoring high-probability messages, we achieve the prediction of embedded messages, thereby enhancing the stability of the optimization process.*"
>
> 2. Regarding your question about the initialized decoding method, we adopt pixel-level comparison for initialized decoding because the embedding of ASIR itself is performed independently between pixels; therefore, pixel-level comparison is a choice aligned with the embedding structure. **It provides a fast and reasonable initial estimation for the subsequent iterative recovery**, allowing the gradient optimization to start from a point closer to the correct one, improving the convergence speed of the optimization process, and shortening the message extraction time. We will supplement this rationale in the revised manuscript to make the logical chain from theoretical motivation to algorithmic implementation more complete.
>
> 3. Regarding embedding timing, we tested different time steps during the preliminary phase. The results on CelebA-HQ are shown below:
>
> | **Time Step** | **Random + Direct** | **Antipodal + Direct** | **Random + Iterative** | **Antipodal + Iterative (ASIR)** |
> | ------------- | ------------------- | ---------------------- | ---------------------- | --------------- |
> | 5th-to-last   | 52.2%               | 53.7%                  | 71.2%                  | **74.6%**       |
> | 4th-to-last   | 52.5%               | 54.0%                  | 77.6%                  | **81.8%**       |
> | 3rd-to-last   | 54.8%               | 59.5%                  | 86.1%                  | **92.5%**       |
> | 2nd-to-last   | 68.7%               | 76.7%                  | 94.6%                  | **99.0%**       |
>
> These results show that embedding at the second-to-last step maximizes the advantages of our scheme (99% accuracy) while avoiding denoising distortion. At earlier steps, basic schemes approach random guessing; **however, at the same early time steps, ASIR can still recover lost signals to the greatest extent and significantly improve extraction accuracy.** Thus, the performance gain is mainly from ASIR itself. We will add these data to the ablation study.
>
> 4. Regarding your question about the payload, we adopt the effective payload as the performance evaluation metric. Under this metric, both ASIR and stegaDDPM achieve an embedding capacity of over 65,536 bits in the pixel domain. Since $65536 / (256 \times 256) = 1$, it reaches the High level ($bpp_{\text{eff}} \ge 1$). In contrast, GRDH only achieves 16,384 bits in the latent space. Since $16384 / (512 \times 512) \approx 0.0625$, it falls in the $0.05 \le bpp_{\text{eff}} < 1$ range, which is categorized as the Medium level.
>
> 5. Thank you for recommending this excellent related work. The research boundary of our paper is strictly focused on **"Training-free"** generative steganography. The mentioned work primarily relies on additional network training. We will supplement the discussion of these training-based methods in the "Related Work" section to clearly define the unique advantages of our training-free route.
>
> 6. To improve readability, we will split Figure 2 into three subfigures focusing on the embedding, initial decoding, and iterative recovery phases. Each subfigure will retain only the core data flow, remove redundant examples, and uniformly use consistent colors to distinguish sender and receiver operations.
>
> Finally, thank you again for your constructive comments. These suggestions play a crucial role in improving our paper, and we will revise accordingly.

---

> > ### Author Rebuttal · Reviewer_pZK3 · 2026-04-06
> >
> > The author has addressed my concerns through detailed explanations and experimental results to a certain extent.

---

> > > ### Author Response · Authors · 2026-04-06
> > >
> > > Dear Reviewer,
> > >
> > > Thank you for taking the time to read through our rebuttal. We were genuinely glad to see that our responses helped clear up your concerns!
> > >
> > > Your initial feedback was incredibly constructive and really pushed us to fix some blind spots in our presentation. **You kindly mentioned earlier that you’d be open to increasing your score if we provided solid clarifications.** We hope our revisions did exactly that! If you feel we've hit the mark, we’d be absolutely thrilled if you might consider updating your score (just as a quick note, the system allows this by editing your original review).
> > >
> > > But honestly, regardless of your final decision, thank you for helping us make this a better paper. We really appreciate the time and energy you've put into reviewing our work.
> > >
> > > Warmly,
> > > The Authors

---

### Decision · Program_Chairs · 2026-04-30

**Decision:**

Accept (regular)

**Comment:**

The paper tackles the signal attenuation and the capacity-robustness trade-off issues in diffusion-based steganography, proposing Antipodal Sampling and Iterative Recovery (ASIR). The reviewers acknowledge the importance of the problem, the clarity of the motivation, and the strong empirical performance across both pixel-space and latent-space settings.

Three out of four reviewers support acceptance at the end of the rebuttal period. While one reviewer assigned a score of 2 and did not update the score, the AC finds that the major concerns raised by this reviewer were largely addressed through the rebuttal.

That said, some concerns remain only partially resolved. In particular, AC finds that the concerns regarding robustness under more realistic or less controlled settings are valid and require further experimental validation. Related issues, such as the reliance on synchronized generation conditions and the limited empirical validation of generalization across architectures and schedulers, also remain only partially addressed.

Overall, the AC agrees that the paper presents a technically solid and well-motivated contribution. However, given the remaining concerns, particularly regarding robustness, AC recommends a weak acceptance at this time; AC also encourages the authors to further strengthen the respective analyses and incorporate them into the revision.